# Pocket warming of bupivacaine with fentanyl to shorten onset of labor epidural analgesia: A double-blind randomized controlled clinical trial

Tyler M. Balon[1‡], Yun Xia[2‡], Johnny McKeown[3], Jack Wang[1], Justin J. Abbott[1], Marilly Palettas[4], Alberto Uribe[2], Marco Echeverria Villalobos[2], John C. Coffman[2], Ling-Qun Hu[2]*

1 College of Medicine, The Ohio State University, Columbus, Ohio, United States of America, 2 Department of Anesthesiology, The Ohio State University Wexner Medical Center, Columbus, Ohio, United States of America, 3 College of Medicine and Life Sciences, The University of Toledo, Toledo, Ohio, United States of America, 4 College of Medicine Center for Biostatistics, The Ohio State University, Columbus, Ohio, United States of America

‡ TMB and YX also contributed equally to this work.
* LingQun.Hu@osumc.edu

**Data Availability Statement:** All relevant data are within the paper and its Supporting Information files.

## Abstract

Shortening analgesic onset has been researched and it has been documented that pre-warming epidural medications to body temperature (37°C) prior to administration increases medication efficacy. Our double-blind randomized controlled trial was designed to investigate if a lower degree of prewarming in providers' pockets could achieve similar results without the need of a bedside incubator. A total of 136 parturients were randomized into either the pocket-warmed group or the room temperature group to receive 10 mL of 0.125% bupivacaine with 2 μg/mL fentanyl epidural bolus at either the 27.8 ±1.7°C or 22.1 ±1.0°C temperatures, respectively. Primary outcome, time to analgesic onset (verbal rating scale pain score ≤ 3) was recorded in 0-, 5-, 10-, 15-, 20-, 30-, and 60-minutes intervals. It was observed that the pocket-warming group (n = 64) and room temperature group (n = 72) had no significant difference of analgesic onset time (median 8 vs. 6.2 minutes; p = 0.322). The incidence of adverse events such as hypotension, fever (≥ 38°C), nausea, vomiting, and number of top-off epidural boluses, as well as patient satisfaction rates and mode of delivery, were not significantly different between the groups as well. Further research is warranted to confirm these findings and explore the impact of different temperatures on analgesic onset time as well as the logistical issues associated with their clinical implementations.

## Introduction

Labor epidural analgesia (LEA) is a widely accepted practice and is a safe and effective method to control pain during labor, usually taking only 10–20 minutes for analgesic onset to occur

**Funding:** This study was supported by the MDSR Roessler Scholarship from The Ohio State University College of Medicine to Tyler M. Balon and Justin J. Abbott, and by the Foundation of Anesthesia Education and Research (FAER)'s Medical Student Anesthesia Research Fellowships to Johnny McKeown.

**Competing interests:** The authors have declared that no competing interests exist.

**Abbreviations:** LEA, Labor epidural analgesia; VRS, Verbal Rating Scale; IQR, interquartile ranges; N, number of participants; %, percentage; CI, Confidence interval; SD, standard deviation; PIEB, programmed intermittent epidural bolus; MDSR, Medical Student Research; FAER, Foundation for Anesthesia Education and Research.

[1]. To achieve more rapid onset of LEA, researchers and physicians have tried strategies such as utilizing faster-acting local anesthetics such as chloropropane, employing higher concentration agents, adding various adjuvants such as sodium bicarbonate, or using other neuraxial approaches such as combined-spinal epidural and dural puncture epidural [2]. Warming the initial dose of epidural medications to 37˚C prior to administration has been studied as well, and it was shown to shorten the onset of action of the medication by more than 5 minutes without lowering its analgesic properties [3]. However, the use of expensive incubators to warm controlled substances in labor and delivery rooms may not be practical in every clinical setting.

Could providers replicate this effect by prewarming the initial epidural doses in their pockets, albeit to a lesser temperature, to achieve a similar result? [4] If so, it would be expected to be less expensive and readily available to every obstetric anesthesia service.

Our null hypothesis is that there is no difference in onset of LEA between patients who receive either a pocket-warmed or room temperature bolus of bupivacaine with fentanyl.

## Materials and methods

This is a prospective, double-blind, randomized clinical trial. The study protocol was approved by approval from our local IRB, Office of Responsible Research Practices (ORRP)—The Ohio State University in 2016 (Protocol ID: 2016H0153) and registered at ClinicalTrails.gov (NCT02912078). The principal investigator was changed on January 26, 2022. Written Informed Consents were obtained in all the study subjects. The primary outcome is onset time of Verbal Rating Scale (VRS) pain score $\leq$ 3 after the initial LEA dose. A total of 62 patients in each group would have 90% power to detect a 5-minute difference, using a one-way analysis of variance with $\alpha = 0.05$. These estimates were based on an effect size of 0.42 ($\mu_1 = 4$, $\mu_2 = 9$, $\sigma = 6$) (NCSS Statistical Software. PASS 2016 (Power Analysis and Sample Size Software). NCSS, LLC, 2016). We intended to include 75 patients in each group (total estimated sample size of 150) to allow for patients that may not obtain adequate labor analgesia at any time after epidural placement and administration of medication. The number of patients was determined based on the sample size calculation as well as a 6.8% incidence of inadequate LEA that was previously reported at an academic center [3, 5]. Inclusion criteria included women requesting LEA with a single vertex presentation fetus at term, with either intact fetal membranes or membrane rupture <6 hours previously. Exclusion criteria included patients with chronic pain, allergies or significant adverse reactions to routinely used local anesthetics or opioids, a contraindication to epidural placement, a baseline temperature over 38˚C, baseline VRS < 3, cervical dilation over 8 cm, or any patient that did not speak English, was incarcerated (to protect their autonomy), or less than 18 years of age.

Our research team was divided into unblinded and blinded personnel. After screening and enrolling patients, unblinded investigators used REDCap to randomize subjects. Block randomization with a block size of 6 was used as the randomization scheme. Both the study subjects and data collecting researchers were blinded from the study groups. A designated researcher collected temperature data alone, was blinded from all other data, and was not present in patients' rooms. An unblinded anesthesiologist, was responsible for randomizing patients, administering medications, measuring temperatures, and subsequently reporting temperature data to the designated unblinded researcher without disclosing any other data. Finally, additional blinded team members were responsible for collecting all other data such as the primary and secondary outcomes. The study was conducted in individual labor and delivery rooms.

Syringes containing 10 mL epidural 0.125% bupivacaine with fentanyl 2 µg/mL were premade, double capped with labeled expiration dates, and stored in the anesthesia workroom of

the labor and delivery unit. The study medications were checked out ahead of time by investigating anesthesiologists and kept in the upper pockets of their surgical scrubs for at least 1 hour together with a 10 mL commercially available saline syringe in clear plastic bags. For the pocket warmed group, the medication temperature was obtained by measuring the accompanying saline syringe. For the room temperature group, medication temperature was recorded from a saline syringe stored on a shelf at the same altitude level in the same room. For both groups, disposable skin temperature sensors (accuracy ± 0.2 Celsius with maximum heating transient time 12 sec, DeRoyal Industries Inc, 200 DeBusk Lane, Powell, TN USA) with GE anesthesia monitors were used to measure temperature immediately following administration of epidural medications, and the peak temperatures were recorded after 30 sec.

After screening, parturients were randomized to receive epidural medications that had either been pocket warmed or stored at room temperature. The onset of adequate labor analgesia, defined as pain VRS ≤ 3, was assessed at 0-, 5-, 10-, 15-, 20-, 30-, and 60-minutes (the final VRS assessment) post initial epidural dose inside of the labor and delivery room. Secondary outcomes included shivering, nausea, vomiting, temperature > 38°C, hypotension, number of top-off epidural boluses, mode of delivery, and overall patient satisfaction assessed in a 0–100 scale via interview and chart reviews within 24 hours of delivery, the end point of the study.

## Statistical analysis

Patient demographic and clinical characteristics were summarized for the two study groups using appropriate descriptive statistics. Means with standard deviations or medians with interquartile ranges (IQR) were used for continuous variables, and frequencies and proportions were used for categorical variables. Comparisons between the room temperature and pocket warming groups included maternal demographics, gestational age, pregnancy history, mode of delivery, analgesic onset, VRS pain scores, and other outcomes. Categorical variables were compared between groups using either a Chi-square test or a Fisher's Exact test, and continuous variables were compared using either a two-sample t-test or a Wilcoxon Rank Sum test. The primary outcome of analgesic onset time was presented as median and IQR and compared between groups using Wilcoxon Rank Sum test. The incidence of VRS score ≤ 3/10 within 15 minutes was summarized using frequencies and proportions and compared between groups using a Chi-square test. Mixed-effects linear regression was used to evaluate differences in VRS pain scores between room temperature and pocket warming groups for the initial 60 minutes after epidural placement. Group and time were included as fixed effects in the model, and patients were included as a random effect to account for the correlation between repeated measures on the same patient. Compound symmetry was used as the covariance structure in the model given there was no correlation among repeated measure observations when evaluating the residual plots. Secondary outcomes (number of top-off epidural bolus, shivering, nausea and/or vomiting, hypotension, fever, and patient satisfaction, and mode of delivery) were compared between the two groups using either a Chi-square test, Fisher's Exact test, two-sample t-test, or a Wilcoxon Rank Sum test, where appropriate. Shapiro-Wilk test was hired to assess normality of the data. A value of P less than 0.05 was used to indicate statistical significance. Statistical analyses and plots were performed using SAS 9.4 (SAS Institute, Inc.; Cary, North Carolina, USA).

## Results

The study was conducted between November 2016 to August 2023. A total of 149 patients were consented and subsequently enrolled in the study, however 13 patients were excluded due to not meeting full inclusion criteria (n = 1), meeting one or more exclusion criteria

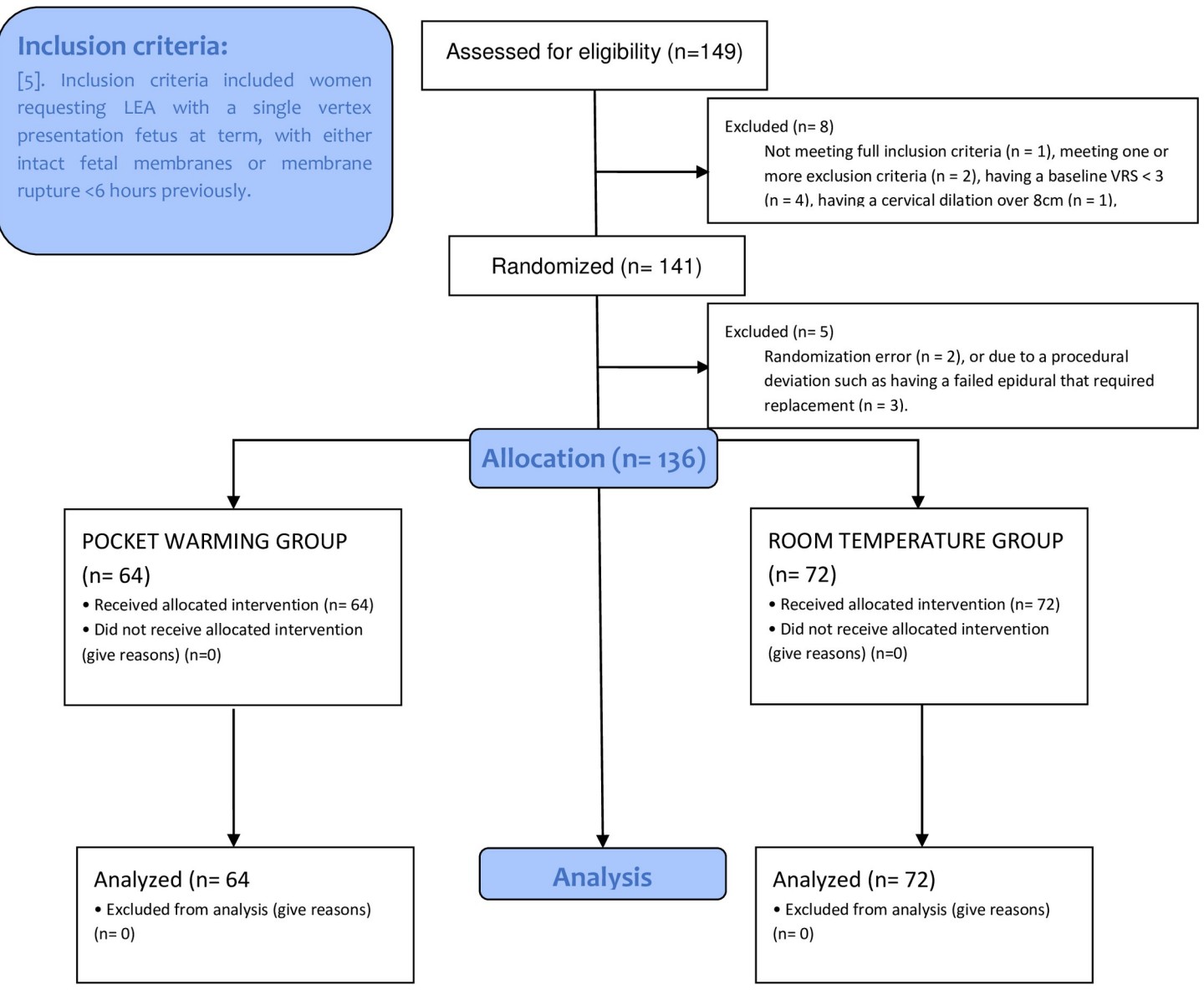

**Fig 1. CONSORT flow diagram for patients enrolled in the study.** n, number of participants; LEA, labor epidural analgesia; VRS: Verbal Rating Score.

(n = 2), having a baseline VRS < 3 (n = 4), having a cervical dilation over 8cm (n = 1), randomization error (n = 2), or due to a procedural deviation such as having a failed epidural that required replacement (n = 3). Therefore, a total of 136 women were analyzed. Of all eligible participants, 64 women were randomized to the pocket warming group, and 72 women were

**Table 1. Demographic summary by group assignment.**

| Variable | Room Temp. (n = 72) | Pocket Warming (n = 64) | P Values |
|---|---|---|---|
| Age (mean (SD)) | 31.6 (5.3) | 30 (4.8) | 0.067 |
| Race | | | |
| White or Caucasian (n (%)) | 55 (76) | 47 (73) | 0.697 |
| Black or African American (n (%)) | 13 (18) | 15 (23) | 0.525 |
| Asian (n (%)) | 3 (4) | 0 (0) | 0.247 |
| Body Mass Index (kg/m$^2$, mean (SD)) | 32.6 (7.1) | 33.1 (6.5) | 0.665 |
| Gestational Age (weeks, mean (SD)) | 39.5 (0.7) | 39.5 (0.8) | 0.971 |
| History of Labor Epidural in Prior Pregnancy, (n (%)) | | | 0.944 |
| Multiparas | 9 (13) | 9 (14) | |
| Nulliparas | 30 (42) | 25 (39) | |
| Gravidity (median [IQR])* | 2 [1, 4] | 2 [1, 2] | 0.023 |
| Cervical Dilation at Initiation of Labor Analgesia (cm, median [IQR])* | 4 [3, 4] | 4 [2.5, 5] | 0.668 |
| Initial Pain Score (VRS, median [IQR])* | 7 [5.8, 8] | 7 [6, 8] | 0.432 |

* the data was non-normal distribution by Shapiro-Wilk test

randomized to the room temperature group. The Consolidated Standards of Reporting Trials (CONSORT) flow diagram is shown in Fig 1 [6].

The two groups were demographically not statistical differences between race, age, BMI, gestational age, previous history of epidurals for laboring pain, cervical dilation at initiation of labor analgesia, and initial VRS pain score prior to epidural administration (**Table 1**). It was determined that the pocket warmed epidural (27.8±1.7˚C) was on average 5.7˚C higher than the room temperature epidural (22.1±1.0˚C). The study medications were observed to reach their maximal temperature following one hour of pocket warming, with any additional duration of warming not being associated with significant increases in medication temperature (P = 0.405, Fig 2).

Legends n, numbers; SD, standard deviation; IQR, interquartile range; VRS, Verbal Rating Score

Shown on Table 2, the two groups were observed to have insignificant difference of analgesic onset when dichotomized by their median time required to reach a VRS score ≤ 3/10 or when dichotomized by a VRS score ≤ 3/10 in the first 15 minutes.

In Mixed Effect Regression Analysis, no difference was detected in pain scores between pocket warming and room temperature groups at any time points after initial bolus: (overall p-value group*time = 0.418) [mean difference (95% CI), p-value—0 min: 0.2 (-0.5, 1.0), P = 0.547; 5 min: 0.6 (-0.1, 1.4), P = 0.112; 10 min: 0.2 (-0.6, 1.0), P = 0.338] (**Fig 3**).

The secondary outcomes (incomplete LEA, shivering, nausea/vomiting, hypotension, fever, patient satisfaction, and mode of delivery) did not differ significantly between the groups (Table 3).

## Discussion

Absence of significantly different analgesic onset time from the initial epidural dose was observed in both the pocket warmed (27.8˚C) and room temperature group (22.1˚C). The results were analyzed with comparisons of median onset time, incidence of VRS score ≤ 3 within 15 minutes, and VRS for pain over time in minutes for the initial 60 minutes after epidural in Mix Effectiveness Analysis. Additionally, secondary outcomes were not statistically significant either.

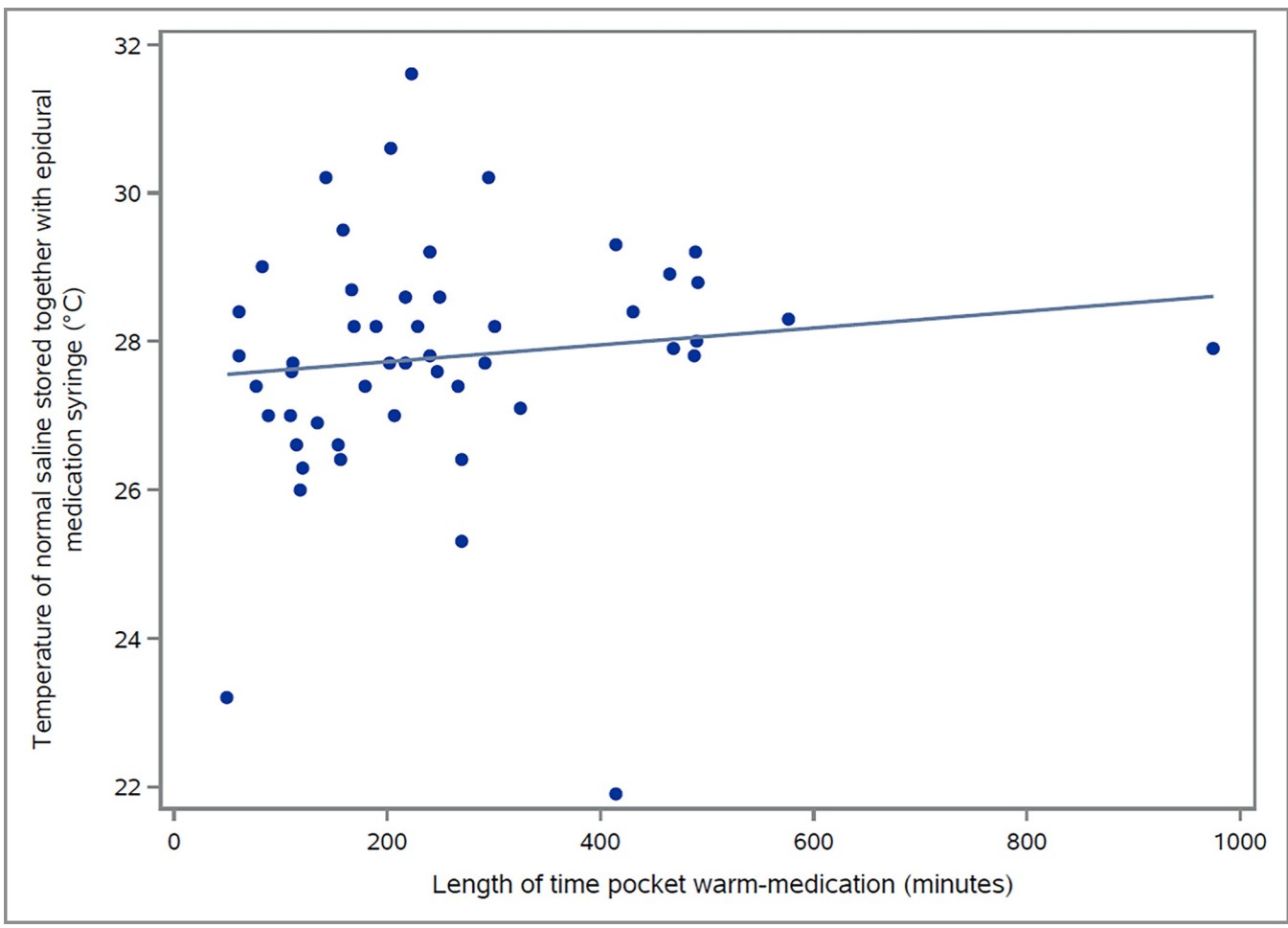

**Fig 2. Association of the temperature of the study medications and the lengths of their pocket warming.** ˚C, Celsius.

These findings differed from the findings in a previous study conducted by Sviggum and et al. [3]. The observed difference in the intervention temperature, 37˚C in the previous study vs. 27.8˚C in this study may play an important role. This is most likely because the temperature of the medication was not warm enough to elicit any significant difference in outcomes. An acceptable temperature, 37˚C, may only be attainable through the use of an incubator rather than through the use of providers' pockets, which only warmed medication to 27.8˚C on average. Notably, pocket warming medications longer than 1 hour did not result in a higher

**Table 2. The primary outcomes between room temperature and pocket warming groups.**

| Variable | Room Temp. (n = 72) | Pocket Warming (n = 64) | P Values |
|---|---|---|---|
| Analgesic onset (minutes) | | | |
| Median [IQR] | 6.2 [4, 15] | 8 [4.1, 16] | 0.322 |
| Mean (SD)* | 10.3 (11.5) | 12.3 (12.4) | |
| VRS score ≤ 3/10 within 15 minutes (n (%)) | 59 (82) | 50 (78) | 0.457 |

*the data was non-normal distribution by Shapiro-Wilk test

Legends: n, numbers; IQR, interquartile range; SD, standard deviation; VRS, Verbal Rating Score

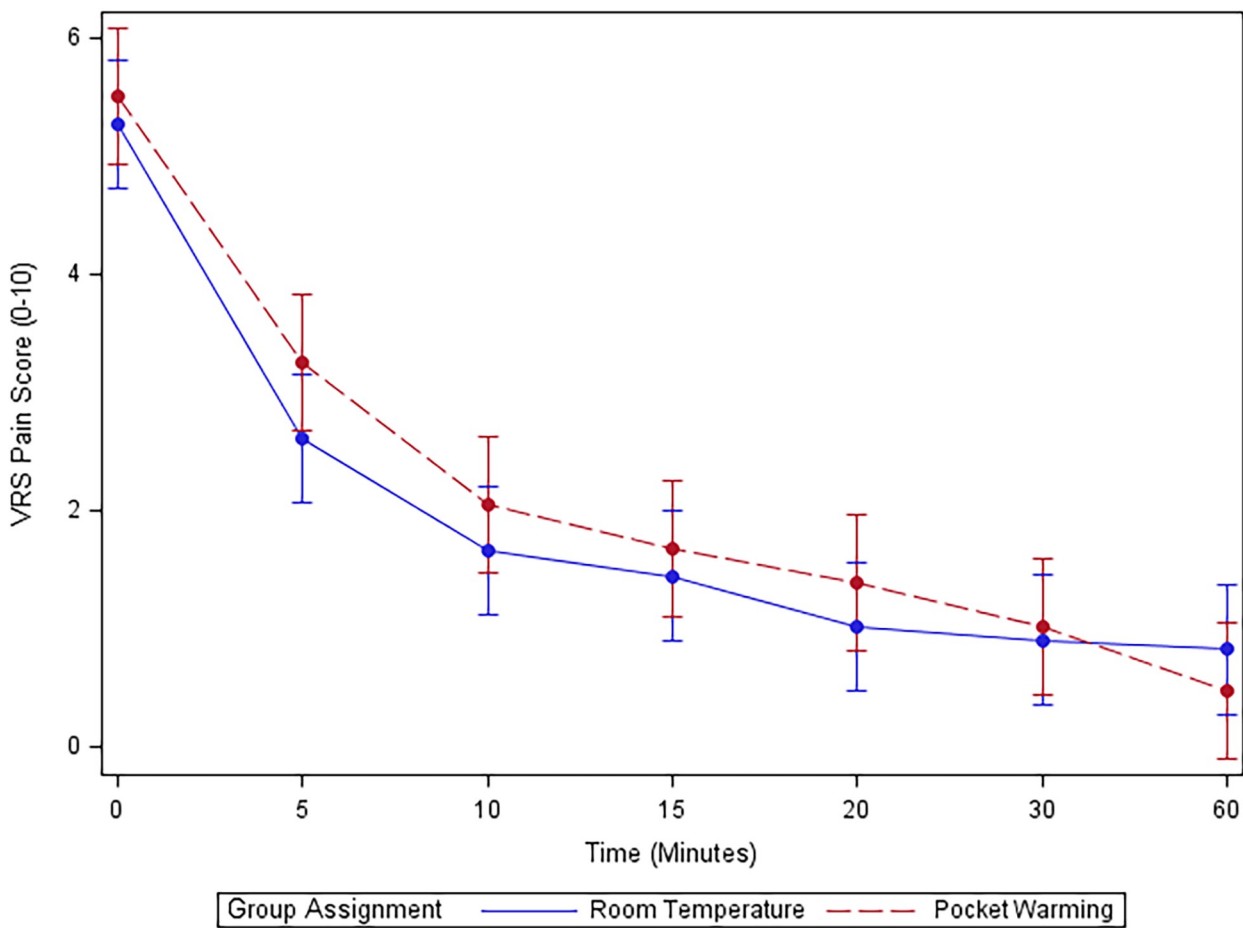

**Fig 3. Verbal rating score for pain over time in minutes for the initial 60 minutes after epidural placement.** VRS, Verbal Rating Score.

**Table 3. The secondary outcomes between room temperature and pocket warming groups.**

| Variable, n (%) | Room Temp. (n = 72) | Pocket Warming (n = 64) | P Values |
|---|---|---|---|
| Top-off epidural bolus (n (%)) | 31 (43) | 23 (36) | 0.397 |
| Shivering (n (%)) | 26 (36) | 20 (31) | 0.55 |
| Nausea and/or Vomiting (n (%)) | 12 (17) | 12 (19) | 0.75 |
| Hypotension (SBP $\leq$ 100, n (%)) | 5 (7) | 4 (6) | 0.871 |
| Fever (Temp >38°C, (n (%)) | 3 (4) | 1 (2) | 0.623 |
| Patient Satisfaction (%, median [IQR]) * | 100 [100, 100] | 100 [100, 100] | 0.903 |
| Mode of delivery (n (%)) | | | 0.692 |
| Cesarean delivery | 16 (22) | 10 (16) | |
| Instrumental delivery | 1 (1) | 1 (2) | |
| Spontaneous vaginal delivery | 55 (76) | 53 (83) | |

*the data was non-normal distribution by Shapiro-Wilk test

Legend: n, numbers; SBP, systolic blood pressure; Temp, temperature; IQR, interquartile range

temperature. This raises the question of what temperature level will result in a significant difference in onset of LEA, and whether the method used to achieve such a temperature could be implemented in a feasible and widely accessible manner.

Likewise, another potential explanation for the difference in results between our findings and those of Sviggum's may be due to the initial dose size. While the same medication was used in both studies, Sviggum used 20mL (as compared to 10mL in this study) which is capable of carrying more heat and would be expected to have a quicker onset time either due to higher drug mass or its heating capacity if the temperature were to play a role [7]. Despite using the same definition for onset time as Sviggum et al., our median analgesic onset time was 6.2 (IQR 4, 15) minutes for the room temperature group, which was quicker than Sviggum's' room temperature onset time of 16.0 ± 10.5 min [3]. In addition, the analgesic onset time in both of our study groups was shorter than 9.2 ± 4.7 min, which was the mean onset time in their prewarmed (37°C) group. It is noted that the analysis methods are different from each other. Time data is usually skewed, especially in small studies. We used and reported median and interquartile description with Wilcoxon Rank Sum test accordingly for comparisons after confirmation of its non-normal distribution with a Shapiro-Wilk test. Interestingly, our mean time ± SD in minutes for the room temperature and pocket warming groups were 10.3 ± 11.5 vs 12.3 ± 12.4, respectively. As such, our routinely used room temperature medications would only take one minute longer of onset than the 37°C solutions if so [3]. It must also be acknowledged that, despite using the same time intervals as were used in previous studies to collect VRS scores, the true LEA onset time is difficult to obtain. Uterine contractions are not continuous and analgesic onset occurs gradually over time rather than at an easily measurable point. Our reporting of the incidence of VRS $\leq$ 3/10 in the first 15 minutes, the cut off based on Sviggum's study, is to attempt to minimize this uncertainty.

Nevertheless, our study did confirm that warming epidural medication to a mean temperature of 27.8°C prior to administration did not pose any harm to parturients with respect to our measured secondary outcomes, i.e. their body temperature, incidences of hypotension, shivering, nausea, vomiting, inadequate LEA, and mode of delivery. This is similar to what has been shown in the previous study [3].

There are several limitations in our study, especially when considering that it took more than 7 years from protocol approval to ultimately finish enrolling patients. During the study period, clinical practice changed over time, one example being that our unit introduced the programmed intermittent epidural bolus (PIEB) pumps in late 2019. Several outcome improvements have been documented after the same PIEB pumps were employed at other institutes [8]. However, the PIEB pump is designed to improve the maintenance of LEA, rather than alter the initiation of LEA which was the primary focus of our study. The primary difference between the two pumps is that the first dose of 10 mL 0.0625% bupivacaine with 2 μg/mL fentanyl starts to be delivered 45 minutes after our initial (study) dose over 5 minutes via the PIEB pumps whereas the same amount of the same medications started to be delivered at the rate of 10 mL/h over 60 minutes evenly via the previous continuous infusion pumps. Patients were not allowed to give themselves patient-controlled epidural analgesia bolus in the study period, i.e. the first 60 minutes. Therefore, this practice change did not change the total amount of medications administered in the study period. Presumably, if it did alter study results, it would occur in both groups evenly without impacts on overall findings. It is also true that our principal investigator changed over time as well as our research team members. The impact of these changes seem very minimal given the fact that the middle trial analysis, mainly based on the data with the old pumps and the previous research team, supported our presumptions [9]. Secondly, we assumed the room temperature was consistently 20°C but it was 22.1 ± 1.0°C by measurement, which resulted in us underestimating the temperature

difference between the two groups. The study sample size calculation was impacted consequently. Therefore, we could not rule out the possibility of type II error for our study.

## Conclusions

In conclusion, the pocket warming (27.8˚C) of bupivacaine with fentanyl did not result in a shorter onset of analgesia when compared with that of room temperature (22.1˚C) bupivacaine with fentanyl. Further study is needed to confirm the previous findings with more commonly used initial dose of bupivacaine with fentanyl or/and practical logistics, especially medication stabilities and their optimal storage period before clinical utilization.

## Supporting information

**S1 Checklist. CONSORT checklist.**
(DOC)

**S1 Data. Pocket warming data for analysis.**
(XLSX)

**S1 File. Pocket warming research protocol.**
(PDF)

## Acknowledgments

The authors gratefully acknowledge Mahmoud Abdel-Rasoul, MS, MPH; Meghan Cook, MD; Harriet Washington, MD; Robert Small, MD; Goran Ristev, MD; Blair H. Hayes, MD; Jason Hoang, BA, McKenna Carr, BS and Jeremy Reeves, BA for their support, writing and editing collaboration that greatly improved our research project and manuscript (without them, it would not be possible to accomplish this project).

## Author Contributions

**Conceptualization:** Yun Xia, Alberto Uribe, John C. Coffman, Ling-Qun Hu.

**Data curation:** Tyler M. Balon, Johnny McKeown, Jack Wang.

**Formal analysis:** Marilly Palettas.

**Funding acquisition:** Tyler M. Balon, Johnny McKeown, Justin J. Abbott, Alberto Uribe, Marco Echeverria Villalobos, Ling-Qun Hu.

**Investigation:** Yun Xia, John C. Coffman, Ling-Qun Hu.

**Methodology:** Marilly Palettas, Alberto Uribe, John C. Coffman, Ling-Qun Hu.

**Project administration:** Yun Xia, Alberto Uribe, Ling-Qun Hu.

**Resources:** Yun Xia, Jack Wang, Alberto Uribe, Marco Echeverria Villalobos, Ling-Qun Hu.

**Software:** Marilly Palettas.

**Supervision:** Yun Xia, Alberto Uribe, Marco Echeverria Villalobos, John C. Coffman, Ling-Qun Hu.

**Validation:** Tyler M. Balon, Johnny McKeown, Jack Wang, Justin J. Abbott, Marilly Palettas, Marco Echeverria Villalobos, Ling-Qun Hu.

**Visualization:** Yun Xia, John C. Coffman, Ling-Qun Hu.

**Writing – original draft:** Tyler M. Balon, Marilly Palettas, Ling-Qun Hu.

**Writing – review & editing:** Yun Xia, Johnny McKeown, Jack Wang, Alberto Uribe, John C. Coffman, Ling-Qun Hu.

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
