## [Decision Letter · Decision Letter 0]

26 Apr 2024

PONE-D-24-05704Pocket Warming of Bupivacaine with Fentanyl to Shorten Onset of Labor Epidural Analgesia: A Double-Blind Randomized Controlled Clinical TrialPLOS ONE

Dear Dr. Hu,

Thank you for submitting your manuscript to PLOS ONE. After careful consideration, we feel that it has merit but does not fully meet PLOS ONE’s publication criteria as it currently stands. Therefore, we invite you to submit a revised version of the manuscript that addresses the points raised during the review process.

We look forward to receiving your revised manuscript.

Kind regards,

Mahmoud Mohammed Alseoudy, M.D.

Guest Editor

PLOS ONE

Journal Requirements:

"1) Medical Student Research (MDSR) Roessler Scholarship from The Ohio State University – College of Medicine: 

Winters: 

Tyler M. Balon: data collection, drafting

Justin J. Abbott: data collection

URL: https://medicine.osu.edu/research/opportunities/mdsr-resources/apply-for-medical-student-research-scholarship

2) The Foundation of Anesthesia Education and Research (FAER)'s Medical Student Anesthesia Research Fellowships

Winner: Johnny McKeown, data collection, drafting;

URL: https://www.asahq.org/faer/programs/medicalstudentfellowships/summerfellowships"

Reviewers' comments:

Reviewer's Responses to Questions

**Comments to the Author**

1. Is the manuscript technically sound, and do the data support the conclusions?

Reviewer #1: Yes

Reviewer #2: Yes

2. Has the statistical analysis been performed appropriately and rigorously? 

Reviewer #1: Yes

Reviewer #2: I Don't Know

3. Have the authors made all data underlying the findings in their manuscript fully available?

Reviewer #1: Yes

Reviewer #2: Yes

4. Is the manuscript presented in an intelligible fashion and written in standard English?

Reviewer #1: Yes

Reviewer #2: Yes

5. Review Comments to the Author

Reviewer #1: A randomized controlled clinical trial was conducted which aimed to investigate if prewarming of epidural medications in providers’ pockets reduced time to analgesic onset compared to the administration of medications at room temperature. The two groups had similar time to analgesic onset and adverse event rates.

Minor revisions:

1- Line 110: Include the full details for the sample size justification. The power calculation should include: (1) the estimated outcomes in each group; (2) the α (type I) error level; (3) the statistical power (or the β (type II) error level); (4) the target sample size, (5) the statistical testing method, and (6) for continuous outcomes, the standard deviation of the measurements.

2- Line 121: If block randomization was used, state the block size.

3- Line 154: Indicate the underlying covariance structure used in the mixed effects models and the criteria for selecting it.

4- The standard statistical term for average is mean.

5- Table 1: Provide only one p-value for comparing race. The one from the overall chi-square or Fisher’s exact test.

6- Line 208: Replace the two instances of the term “stratified” because it has a specific statistical meaning in relation to the randomization process which has not been conducted in the study. Perhaps “dichotomized” is a more descriptive term.

7- Line 216: Provide the overall p-value for testing the interaction effect of group by time. If the interaction effect is significant, provide an interpretation of the results, but do not test main effects because the tests for main effects are uninteresting in light of significant interactions. If interaction effects are non-significant, drop the interaction effects from the model and test the main effects. Determining which results to present when testing interactions is often a multi-step process.

8- Line 223: For clarification, indicate that the secondary outcomes (rate of adverse events) did not differ significantly between the groups.

Reviewer #2: I would like to comment on the long period of duration and unnecessary delay.

The anesthesiologist was not blind and this may expose the trial to selection bias.

I would prefer that there was a control group with no warming.

6. PLOS authors have the option to publish the peer review history of their article (what does this mean?). If published, this will include your full peer review and any attached files.

Reviewer #1: No

Reviewer #2: No

---

## [Author Response · Author response to Decision Letter 0]

12 May 2024

Minor revisions:

1- Line 110: Include the full details for the sample size justification. The power calculation should include: (1) the estimated outcomes in each group; (2) the α (type I) error level; (3) the statistical power (or the β (type II) error level); (4) the target sample size, (5) the statistical testing method, and (6) for continuous outcomes, the standard deviation of the measurements.

Addressed in the manuscript. “A total of 62 patients in each group would have 90% power to detect a 5-minute difference, using a one-way analysis of variance with α=0.05. These estimates were based on an effect size of 0.42 (µ1=4, µ2=9, σ=6). We intend to include 75 patients in each group (total estimated sample size of 150) to allow for patients that may not obtain adequate labor analgesia at any time after epidural placement and administration of medication. The number of patients was determined based on the sample size calculation as well as a 6.8% incidence of inadequate LEA that was previously reported at an academic center.”

2- Line 121: If block randomization was used, state the block size.

Addressed in the manuscript – block size of 6 was used.

3- Line 154: Indicate the underlying covariance structure used in the mixed effects models and the criteria for selecting it.

Addressed in the manuscript – compound symmetry was the covariance structure used in the mixed effects model. No particular pattern in the correlation among repeated measure observations was seen when evaluating the residual plots.

4- The standard statistical term for average is mean.

Addressed in the manuscript.

5- Table 1: Provide only one p-value for comparing race. The one from the overall chi-square or Fisher’s exact test.

Each category of race was captured as a separate variable and the proportions and p-values presented are the “Yes” vs “No” comparisons for each race category.

6- Line 208: Replace the two instances of the term “stratified” because it has a specific statistical meaning in relation to the randomization process which has not been conducted in the study. Perhaps “dichotomized” is a more descriptive term.

Addressed in the manuscript.

7- Line 216: Provide the overall p-value for testing the interaction effect of group by time. If the interaction effect is significant, provide an interpretation of the results, but do not test main effects because the tests for main effects are uninteresting in light of significant interactions. If interaction effects are non-significant, drop the interaction effects from the model and test the main effects. Determining which results to present when testing interactions is often a multi-step process.

Addressed in the manuscript – The overall p-value for the interaction effect of group by time was 0.418 and it was dropped from the model. Time was the only significant main effect (p<0.001).

8- Line 223: For clarification, indicate that the secondary outcomes (rate of adverse events) did not differ significantly between the groups.

Addressed in the manuscript.

---

## [Decision Letter · Decision Letter 1]

23 Jul 2024

PONE-D-24-05704R1Pocket Warming of Bupivacaine with Fentanyl to Shorten Onset of Labor Epidural Analgesia: A Double-Blind Randomized Controlled Clinical TrialPLOS ONE

Dear Dr. Hu,

Thank you for submitting your manuscript to PLOS ONE. After careful consideration, we feel that it has merit but does not fully meet PLOS ONE’s publication criteria as it currently stands. Therefore, we invite you to submit a revised version of the manuscript that addresses the points raised during the review process.

**ACADEMIC EDITOR: Please respond to all reviewers comments**

We look forward to receiving your revised manuscript.

Kind regards,

Ahmed Mohamed Maged, MD

Academic Editor

PLOS ONE

Journal Requirements:

Reviewers' comments:

Reviewer's Responses to Questions

**Comments to the Author**

1. If the authors have adequately addressed your comments raised in a previous round of review and you feel that this manuscript is now acceptable for publication, you may indicate that here to bypass the “Comments to the Author” section, enter your conflict of interest statement in the “Confidential to Editor” section, and submit your "Accept" recommendation.

Reviewer #1: All comments have been addressed

Reviewer #3: All comments have been addressed

Reviewer #4: All comments have been addressed

Reviewer #5: All comments have been addressed

Reviewer #6: All comments have been addressed

2. Is the manuscript technically sound, and do the data support the conclusions?

Reviewer #1: (No Response)

Reviewer #3: Yes

Reviewer #4: Partly

Reviewer #5: (No Response)

Reviewer #6: Partly

3. Has the statistical analysis been performed appropriately and rigorously? 

Reviewer #1: (No Response)

Reviewer #3: Yes

Reviewer #4: N/A

Reviewer #5: (No Response)

Reviewer #6: I Don't Know

4. Have the authors made all data underlying the findings in their manuscript fully available?

Reviewer #1: (No Response)

Reviewer #3: Yes

Reviewer #4: Yes

Reviewer #5: (No Response)

Reviewer #6: No

5. Is the manuscript presented in an intelligible fashion and written in standard English?

Reviewer #1: (No Response)

Reviewer #3: Yes

Reviewer #4: Yes

Reviewer #5: (No Response)

Reviewer #6: Yes

6. Review Comments to the Author

Reviewer #1: (No Response)

Reviewer #3: To authors,

First of all, I want to thank authors for this interesting review and for their efforts.

Title: Well written.

Abstract: well written

Background: is well written.

Methods: Thanks to authors as methods is well written in details.

Results:

• The data is statistically analyzed and expressed well.

• Tables and figures are expressed the data well.

Discussion: well written and relevant to the results of the study.

Conclusion: expressed the finding well and well written.

Best regard

Reviewer

Reviewer #4: i acknowledge the authors efforts in conducting this long duration trial. the subject is interested and of clinical importance which was conducted to explore if pocket warming would achieve similar results as body temperature warmed medications as compared to room temperature medications, however i have the following comments:

1- first it is better to describe your results ( analgesia onset) by absence of significant difference between both groups and not describing them as similar results because there was an observed difference in favor of room temperature group. - please correct it in abstract, results and discussion.

2- regarding the study outcomes, they needs more clarification regarding their definition- time points- end time point- method of assessment. for example the onset time was mentioned to be assessed for 60 minutes which doesn't reflect the clinical practice. onset of block is usually assessed every 5min within 30 min after the block if no adequate response extra-dose will be added. the time-point (60min) for follow up of the VRS score not the onset only.

and what did authors do if the analgesia target was not achieved .. they didn't exclude such cases.

when and how the rescue epidural doses were given? and why this outcome not reported in the results!

how did you assess the patient satisfaction?

are skin temperature sensors valid to measure the syringe temperature? mention your reference

3- as regard sample size, mention the used software to calculate the sample size,

the reference of the effect size, and the reference of The minimum clinically important difference (5min)

4- as regard the statistical analysis; mention the used test of normality

how did you perform the subgroup analysis in the pocket warming group to evaluate the durations of warming as reported in results

5- in Results; in CONSORT chart, if authors performed allocation before giving the epidural, so failed procedure should be inserted in the drop-out cases that didn't receive the allocated intervention.

illustrate the method of the duration of pocket warming and patient satisfaction and how they were presented in results

please revise the attached file.

Reviewer #5: (No Response)

Reviewer #6: Thank you for giving me the opportunity to review this interesting manuscript.

This study investigated whether prewarming epidural medications in providers' pockets (to approximately 27.8°C) can shorten the onset time of labor epidural analgesia compared to room temperature medications (22.1°C). The primary outcome, median time to analgesic onset, and secondary outcomes, including incidences of adverse events such as hypotension, fever, nausea, vomiting, the number of rescue epidural boluses, and patient satisfaction rates, showed no significant difference between the groups.

It was interesting but I have some questions.

One of the most important methodological flaws of this study is the inconsistent temperature measurement of bupivacaine.

The study relied on indirect methods to estimate the temperature of the epidural solution. The temperature of the accompanying saline syringe was used to approximate the temperature of the bupivacaine solution. Direct measurement of the medication temperature immediately before administration would have provided more accurate and reliable data. This inconsistency may have influenced the study outcomes and should be addressed for greater accuracy.

Please discuss how the temperature was consistently maintained during the pocket warming and any potential variations observed.

The manuscript also does not provide detailed information specifically describing the aseptic methods used during the storage of solutions in the pocket. It is critical to ensure that proper aseptic techniques are followed to prevent contamination and infection. A detailed description of these methods should be included to strengthen the study’s validity.

It is known that higher temperatures can cause changes in the density and viscosity of bupivacaine, which in turn affects its cephalad spread compared to cerebrospinal fluid (CSF). Studies such as Anesth Analg. 2006 Jan;102(1):272-5 have shown that a decrease in the density and viscosity of bupivacaine relative to CSF can impact its spread. This manuscript should discuss how these temperature-induced changes might influence the onset time of epidural analgesia, which is not administered intrathecally. Understanding these effects can provide insights into the observed results and help refine the study design.

By addressing these points, the manuscript will provide a more comprehensive and accurate evaluation of the impact of prewarming bupivacaine on the onset time of labor epidural analgesia.

7. PLOS authors have the option to publish the peer review history of their article (what does this mean?). If published, this will include your full peer review and any attached files.

Reviewer #1: No

Reviewer #3: **Yes: **Alshaimaa Abdel Fattah Kamel

Reviewer #4: No

Reviewer #5: No

Reviewer #6: No

---

## [Author Response · Author response to Decision Letter 1]

5 Aug 2024

Dear Dr. Maged,

I am writing to address your comments and those of the reviewers, particularly Reviewer 4 and Reviewer 6, as detailed as attached file titled "Rebuttal letter for Reviewer Comments_31July2024.docx". All the line information is based on the marked version.

Additionally, I have reorganized the parameter "mode of delivery" from Table 1 to Table 3 and have updated the corresponding text for better presentation.

For clarity, our responses are prefixed with "RESPONSE" and highlighted in purple.

Please let us know if you have any further questions or comments.

Sincerely,

Ling-Qun Hu, MD

---

## [Decision Letter · Decision Letter 2]

14 Aug 2024

Pocket Warming of Bupivacaine with Fentanyl to Shorten Onset of Labor Epidural Analgesia: A Double-Blind Randomized Controlled Clinical Trial

PONE-D-24-05704R2

Dear Dr. Hu,

We’re pleased to inform you that your manuscript has been judged scientifically suitable for publication and will be formally accepted for publication once it meets all outstanding technical requirements.

Kind regards,

Ahmed Mohamed Maged, MD

Academic Editor

PLOS ONE

Additional Editor Comments (optional):

Reviewers' comments:

Reviewer's Responses to Questions

**Comments to the Author**

1. If the authors have adequately addressed your comments raised in a previous round of review and you feel that this manuscript is now acceptable for publication, you may indicate that here to bypass the “Comments to the Author” section, enter your conflict of interest statement in the “Confidential to Editor” section, and submit your "Accept" recommendation.

Reviewer #4: All comments have been addressed

Reviewer #6: (No Response)

2. Is the manuscript technically sound, and do the data support the conclusions?

Reviewer #4: Yes

Reviewer #6: Yes

3. Has the statistical analysis been performed appropriately and rigorously? 

Reviewer #4: Yes

Reviewer #6: Yes

4. Have the authors made all data underlying the findings in their manuscript fully available?

Reviewer #4: Yes

Reviewer #6: Yes

5. Is the manuscript presented in an intelligible fashion and written in standard English?

Reviewer #4: Yes

Reviewer #6: Yes

6. Review Comments to the Author

Reviewer #4: the revised version of the manuscript is much improved.

The authors have addressed all my comments for this paper and answered the technical questions I have for this method. Therefore, I have no further comments.

Reviewer #6: Thank you for addressing the response regarding the lack of references and limited data on the impact of local anesthetic temperature on epidural onset. While I acknowledge the authors’ point that there is limited direct evidence on this topic within the context of epidural analgesia, I believe it is crucial to explore and discuss this aspect further. The Braz J Anesthesiol (2021) study provides a valuable perspective by suggesting that temperature changes may influence not only the intrathecal spread of anesthetics but also their direct action on spinal nerve roots.

Previous studies have proposed hypotheses that could be relevant to the current investigation. For instance, increasing the temperature of a local anesthetic decreases its dissociation constant, which in turn increases the unionized fraction of the drug. This change enhances lipid solubility, potentially leading to greater membrane permeation and more effective nerve blockade. Furthermore, the vasoconstrictive effects of cooling, which reduce the amplitude and increase the duration and latency of action potentials, could explain why cooling might delay onset. Conversely, warming could mitigate these effects, promoting faster recovery of inactivated fibers and potentially accelerating the onset of anesthesia.

Given these considerations, I recommend that the authors refer to studies such as Rosenberg and Heavner (1980), Sviggum et al. (2015), Kamaya et al. (1983), Lee et al. (2012), and Pappone (1980) to support a more detailed hypothesis. If space constraints prevent a full discussion within the main text, these points should at least be acknowledged in the limitations section to provide a more comprehensive understanding of the study’s context and the potential physiological implications of temperature variations on local anesthetic efficacy. Addressing this could significantly strengthen the manuscript by connecting existing knowledge with the current study’s findings and offering a more nuanced interpretation of the results.

1. Rosenberg P, Heavner J. Temperature-dependent nerve-blocking action of lidocaine and halothane. Acta Anaesthesiol Scand. 1980;24:314-320.

2. Sviggum H, Yacoubian S, Liu X, et al. The effect of bupivacaine with fentanyl temperature on initiation and maintenance of labor epidural analgesia: a randomized controlled study. Int J Obstet Anesth. 2015;24:15-21.

3. Kamaya H, Hayes JJ, Ueda I. Dissociation constants of local anesthetics and their temperature dependence. Anesth Analg. 1983;62:1025-1030.

4. Lee R, Kim YM, Choi EM, et al. Effect of warmed ropivacaine solution on onset and duration of axillary block. Korean J Anesthesiol. 2012;62:52-56.

5. Pappone PA. Voltage-clamp experiments in normal and denervated mammalian skeletal muscle fibres. J Physiol. 1980;306:377-410.

7. PLOS authors have the option to publish the peer review history of their article (what does this mean?). If published, this will include your full peer review and any attached files.

Reviewer #4: No

Reviewer #6: **Yes: **Ki Tae Jung

---

## [Editor Report · Acceptance letter]

23 Aug 2024

PONE-D-24-05704R2 

PLOS ONE

Dear Dr. Hu, 

I'm pleased to inform you that your manuscript has been deemed suitable for publication in PLOS ONE. Congratulations! Your manuscript is now being handed over to our production team.

Kind regards, 

on behalf of

Professor Ahmed Mohamed Maged 

Academic Editor

PLOS ONE